# Cardiomyocyte-Restricted Expression of IL11 Causes Cardiac Fibrosis, Inflammation, and Dysfunction

**DOI:** 10.3390/ijms241612989

**Published:** 2023-08-20

**Authors:** Mark Sweeney, Katie O’Fee, Chelsie Villanueva-Hayes, Ekhlas Rahman, Michael Lee, Konstantinos Vanezis, Ivan Andrew, Wei-Wen Lim, Anissa Widjaja, Paul J. R. Barton, Stuart A. Cook

**Affiliations:** 11MRC-London Institute of Medical Sciences, Hammersmith Hospital Campus, London W12 0NN, UK; 2Institute of Clinical Sciences, Faculty of Medicine, Imperial College, London W12 0NN, UK; 3Wellcome Trust/NIHR 4i Clinical Research Fellow, Imperial College, London W12 0NN, UK; 4National Heart and Lung Institute, Imperial College, London W12 0NN, UK; 5National Heart Research Institute Singapore, National Heart Centre Singapore, Singapore 169609, Singapore; 6Cardiovascular and Metabolic Disorders Program, Duke-National University of Singapore Medical School, Singapore 169857, Singapore; 7Royal Brompton and Harefield Hospitals, Guy’s and St. Thomas’ NHS Foundation Trust, London SW3 6NP, UK

**Keywords:** Interleukin-11, fibrosis, EndMT, heart failure

## Abstract

Cardiac fibrosis is a common pathological process in heart disease, representing a therapeutic target. Transforming growth factor β (TGFβ) is the canonical driver of cardiac fibrosis and was recently shown to be dependent on interleukin 11 (IL11) for its profibrotic effects in fibroblasts. In the opposite direction, recombinant human IL11 has been reported as anti-fibrotic and anti-inflammatory in the mouse heart. In this study, we determined the effects of IL11 expression in cardiomyocytes on cardiac pathobiology and function. We used the Cre-loxP system to generate a tamoxifen-inducible mouse with cardiomyocyte-restricted murine *Il11* expression. Using protein assays, bulk RNA-sequencing, and in vivo imaging, we analyzed the effects of IL11 on myocardial fibrosis, inflammation, and cardiac function, challenging previous reports suggesting the cardioprotective potential of IL11. TGFβ stimulation of cardiomyocytes caused *Il11* upregulation. Compared to wild-type controls, *Il11*-expressing hearts demonstrated severe cardiac fibrosis and inflammation that was associated with the upregulation of cytokines, chemokines, complement factors, and increased inflammatory cells. IL11 expression also activated a program of endothelial-to-mesenchymal transition and resulted in left ventricular dysfunction. Our data define species-matched IL11 as strongly profibrotic and proinflammatory when secreted from cardiomyocytes and further establish IL11 as a disease factor.

## 1. Introduction

Interleukin 11 (IL11) is a little-studied and poorly understood member of the interleukin 6 family of cytokines. Initial reports on the role of IL11 in the myocardium described cardioprotective, anti-fibrotic, and anti-inflammatory effects of recombinant human IL11 (rhIL11) in the setting of myocardial ischemia in mice [1]. Similarly, studies in other organ systems have shown the anti-inflammatory effects of rhIL11 in other mouse models of disease [2]. However, a major limitation of these early studies was the systematic use of non-species-matched rhIL11 in mice, and it was subsequently shown that rhIL11 has unexpected effects in mice [3]. More recently, a range of loss- and (species-matched) gain-of-function studies have shown a profibrotic [4,5,6,7] and proinflammatory effect of IL11 in various contexts [8,9,10].

IL11 is not easily detected in healthy tissues but is expressed in parenchymal, mesenchymal, and epithelial cells as an alarmin-like response following exposure to a range of cell stressors such as viral, toxin, bacterial, inflammatory, mechanical, oxidative, and genetic factors [11]. In particular, transforming growth factor β (TGFβ) strongly upregulates IL11 in human and mouse cardiac fibroblasts [5]. IL11 expression is increased in mouse models of cardiac disease including transverse aortic constriction, angiotensin II infusion [12], and ischemia [1]. In human and non-human primate studies, exogenous IL11 causes heart failure symptoms, myocardial hypertrophy, and elevation of serum B-natriuretic peptide levels. [13,14] Serum levels of IL-11 are associated with heart failure symptoms, atrial fibrosis, and atrial fibrillation in humans [15,16].

To date, the effects of IL11 in the heart have focused on its role in fibroblasts using organismal-level loss- and gain-of-function studies [5], and the effect of IL11 secretion from cardiomyocytes is not known. Equally, the true activity of IL11 in cardiac inflammation remains poorly characterized and controversial [1,5,17]. In this study, we established a new transgenic mouse model with temporally regulated and cardiomyocyte-specific *Il11* expression to understand the effects of IL11 secretion from non-mesenchymal cells in the adult heart.

## 2. Results

### 2.1. IL11 Is Expressed by Cardiomyocytes

TGFβ plays an important role in cardiac pathology following MI and is known to stimulate IL11 secretion from fibroblasts and epithelial cells [5,18]. We therefore tested if cardiomyocytes upregulate *Il11* in response to TGFβ stimulation using primary adult mouse cardiomyocytes. Compared to untreated cells, TGFβ stimulation (5 ng/mL, 24 h) induced a seven-fold increase in the expression of *Il11* (Figure 1a,b). To determine the effects of IL11 secretion by cardiomyocytes in adults in vivo, we generated a transgenic mouse model (CM-Il11-Tg). The Cre-loxP system, using the *Rosa26-Il11* transgenic mouse, was used as previously described [5,19]. This mouse was crossed with an α-MHC-MerCreMer (MCM) mouse [20] (Figure 1c–e) to create cardiomyocyte-specific expression of *Il11* in response to tamoxifen administration, subsequently referred to as CM-Il11-Tg (Appendix A).

### 2.2. IL11 Secreted by Cardiomyocytes Has Paracrine Profibrotic Effects

Following tamoxifen administration, *Il11* was upregulated in the hearts of CM-Il11-Tg mice (Figure 2a and Appendix A). QPCR analysis of myocardial extracts six weeks after tamoxifen administration showed large upregulation of profibrotic (*Col1a1*, *Col3a1*, *Fn1*, and *Postn*) and extracellular matrix remodeling (*Mmp2*, *Mmp9*, *Mmp14*, and *Timp1*) genes (Figure 2b,c). Masson’s trichrome staining of histological sections revealed a large increase in interstitial, epicardial, and, most notably, perivascular fibrosis (Figure 2d–f). These data suggest that the paracrine activity of *Il11* from cardiomyocytes on fibroblasts that causes cardiac fibrosis is notably different to the effects of cardiomyocyte-restricted TGFβ expression, which does not cause fibrosis of the ventricles [21].

### 2.3. Proinflammatory Effects of IL11 in the Heart

RNASeq of left ventricular tissue from male and female mice was performed at 1, 3, and 6 weeks following induction of *Il11* expression and compared to control animals (n = 4 per time point per sex). Gene ontology term analysis of CM-Il11-Tg revealed that the top 20 most differentially regulated gene ontology biological processes (GOBP) terms were related to inflammation, immune response, and leukocyte activation (Figure 3a,b). These changes were consistent across time points and reproducible in both male and female mice (Appendix A). Hallmark gene set enrichment analysis (GSEA) also highlighted multiple inflammatory terms in significantly enriched gene sets, including IL6 JAK-STAT3 signaling, inflammatory response, interferon-gamma response, TNFα signalling via NF-𝜅B, and complement activation (Figure 3d and Appendix A).

Hierarchical clustering of differentially expressed genes from the most enriched GOBP term “immune system process” (GO:0002376) demonstrates that the clusters of gene expression vary across the individual time points. In particular, the genes associated with major histocompatibility complexes are significantly upregulated at 3 and 6 weeks compared to week 1 (*H2-Ab1, H2-DMb1, H2-Eb1*), which is suggestive of infiltration and activation of phagocytic antigen-presenting cells into the myocardium. Additional genes were upregulated at later time points, including factors within the complement pathway (*C4a, C4b,* and *C6*), immunoglobulin-related genes (*Igf2, Ighg2c,* and *Jchain*), lymphocyte markers, and chemotactic signaling molecules (*Cd4, Cd3e,* and *Cxcl13*). A smaller cluster of genes was most strongly upregulated at the one-week time point, including chemokines involved in the recruitment and activation of granulocytes and monocytes (*Il6, Tnf, Cxcl1, Cxcl5, Ccl4, and Ccl7*) (Figure 3c).

The KEGG pathway analysis was consistent with the GOBP analysis and included complement activation, phagosome activity, chemokine signaling pathways, and the NF-𝜅B signaling pathway among the most significantly enriched pathways (Figure 3e and Appendix A). Immunohistochemical staining for macrophages and increased *Cd68* expression in RNAseq supports the development of a proinflammatory environment within the myocardium of CM-Il11-Tg mice, with a focus on the inflammatory changes found predominately around the perivascular area (Figure 3f and Appendix A).

### 2.4. Endothelial-to-Mesenchymal Transition in IL11-Expressing Mice

Beyond fibrosis and inflammation, IL11 has been implicated in epithelial-to-mesenchymal transition (EMT) and endothelial-to-mesenchymal transition (EndMT), which may be of relevance to the strong perivascular phenotype we observed following *Il11* expression (Figure 2d) [6,22]. The EMT Hallmark geneset was in the top six most-enriched genesets following transgene recombination across all time points in male and female mice (Figure 3d,g and Appendix A). Reciprocal to upregulation of EMT genes and mesenchymal genes in CM-Il11-Tg mice, endothelial markers (*Cdh5, Kdr, Flt1, Tie1,* and *Nos3*) were downregulated (Figure 3h). Phosphorylation of STAT3, part of the canonical signaling pathway of IL11, was increased at all time points (Figure 3i,j), and TWIST, a key transcription factor in EndMT and EMT, was upregulated at the protein level at later time points in the CM-Il11-Tg mice (Figure 3i,k).

### 2.5. Impaired Ventricular Function following Cardiomyocyte IL11 Expression

Cardiac function was assessed using transthoracic echocardiography 6 weeks after transgene induction. Left ventricular ejection fraction (LVEF) measures the proportion of blood ejected from the heart during each cardiac contraction. This measure of systolic function was similarly impaired in both male and female CM-Il11-Tg mice compared to tamoxifen-injected controls (Figure 4a,b), suggesting the absence of sexual dimorphism of cardiomyocyte-specific *Il11* expression in heart function. This was accompanied by dilation of the left ventricular cavity, indicating adverse remodeling of the ventricle. Myocardial shortening, a complementary measure of the systolic force-generating capacity of the ventricle, was also reduced when measured using global circumferential strain (Figure 4c,d). Systolic dysfunction was observed within 3 weeks of transgene induction, with differences in diastolic LV size apparent only at the later 6-week time point (Appendix A). In keeping with this early effect on cardiac function, the cardiomyocyte stress markers *Nppb* and *Nppa* were significantly upregulated in the RNA-seq data by the 1-week timepoint in the CM-Il11-Tg mice compared to the WT mice (Appendix A).

## 3. Discussion

Until recently, IL11 was thought to be anti-fibrotic in the heart and other organs and is still thought by many to be anti-inflammatory, although these accepted concepts have been challenged by studies in recent years [2,3,4,5,6,8,9,10,23,24]. Here, we examined the specific effect of IL11 secreted from cardiomyocytes, which were shown to express *Il11* in response to TGFβ stimulation. While TGFβ is viewed as the strongest profibrotic factor, it was striking that cardiomyocyte-restricted *Il11* expression caused robust ventricular fibrosis, whereas similar expression of TGFβ in the same experimental system does not [21]. This difference may reflect the proinflammatory effects of IL11 in a maladaptive loop of fibro-inflammation, whereas TGFβ is anti-inflammatory [25,26,27]. The data may also reflect that the paracrine effects of cardiomyocyte-derived IL11 on neighboring fibroblasts are more pronounced than those of TGFβ, which might be more important as an autocrine factor in fibroblasts.

In our studies, the cardiac inflammation observed following *Il11* induction in mouse cardiomyocytes evolved over time, with an initial cytokine signature and a later footprint of chemokines and complement factors. These effects were profound and reflected, in part, the recruitment of neutrophils, leukocytes, and monocytes, which were apparent in the MAC2 staining. While a review in 2022 [2] speculated that the jury was still out as to whether IL11 was pro- or anti-inflammatory, we believe these data, taken together with other recent publications, strongly support the idea that IL11 is proinflammatory [2,24]. Our data also show that IL11 causes cardiac dysfunction, as has been seen in response to chronic recombinant IL11 injection in mice and IL11 expression in the fibroblast compartment, which further reinforces the negative impact of IL11 on cardiac function [5].

We noted that IL11 expression induced extensive changes within the perivascular region of the myocardium and that this was associated with reduced expression of endothelial markers and upregulation of EndMT genes. Intriguingly, IL11 was found to play a role in EndMT in the pulmonary arteries in the setting of idiopathic pulmonary fibrosis-related pulmonary hypertension [6]. We propose that IL11-driven EndMT directly or indirectly contributes to the perivascular fibrosis demonstrated here and in pulmonary hypertension. Our findings of IL11-associated EMT/EndMT are in line with recent studies of kidney disease, where IL11 causes pathogenic EMT [4].

This study has limitations. It is likely that the tissue levels of IL11 produced following transgene recombination were higher than those seen in the pathology. However, this expression system provides valuable insights into the cardiac biology associated with gene gain-of-function and has been used extensively, including previously to study the effects of TGFβ on cardiac fibrosis, which were minimal. The higher level of expression or more chronic exposure to IL11 in this model may, in part, be responsible for differences in inflammatory response, compared to earlier mouse studies which had concluded an anti-inflammatory role of rhIL11. These studies administered rhIL11 to mice through daily injections and reported reduced pancreatic islet destruction in type 1 diabetes models, reduced synovitis in rheumatoid arthritis models, and reduced colitis in an inflammatory bowel disease model [28,29,30]. The doses of rhIL11 used in these studies were variable; however, in some cases, very large doses of up to 100 mcg per mouse were used, making the possibility of a differential dose effect less likely. The duration of administration of rhIL11 in these studies was between 5 and 10 days of daily injections, which is comparable to the 7-day RNAseq time point in our study, by which point extensive inflammatory transcriptional changes are already present. Another important point to consider is that rhIL11 has unexpected and paradoxical effects compared to rmIL11 when used in mice, and experimental results arising from the use of these different reagents in mouse models should not be directly compared [3].

The endMT changes that we observed did not exclude the possibility that endothelial cell signatures were reduced due to endothelial cell loss from another mechanism, and we did not measure changes in the capillary density in this study; however, the concurrent expression of pro-EMT transcription factors strengthens our interpretation.

We suggest that myocardial stress signals that occur due to ischaemia or pressure overload [1,12] result in the secretion of IL11 from cardiomyocytes, leading to cardiac fibrosis and inflammation. Cardiac fibrosis remains an attractive therapeutic target, but unfortunately, inhibition of TGFβ failed in the clinic due to on-target toxicities, which include inflammation and cardiac impairment [31,32]. IL11 may therefore represent an alternative target, as its effects on cardiac fibrosis appear more profound than TGFβ, and IL11 is also proinflammatory. IL11 has important physiological roles in some aspects of facial and skull bone development and fertility in mice. Female *Il11ra1* knockout animals are infertile, although female humans with loss-of-function in IL11 signaling appear to be fertile [33]. Craniosynostosis is seen in up to 40% of *Il11ra1* knockout mice, which has also been reported in humans with homozygous *IL11RA* loss-of-function mutations [34,35]. Mice with deleted *Il11* have no reported bony abnormalities, and the females are infertile [36]. Mice with selective loss of IL11 signaling due to an R279Q mutation of gp130 have normal cranial sutures and normal bone parameters but show incompletely penetrant snout abnormalities [37]. In adult mice, *Il11* or *Il11ra1* knockout or long-term anti-IL11 antibody experiments do not appear to cause major health issues, indicating a favorable safety profile [38]. This contrasts with genetic or pharmacologic inhibition of TGFβ, which is detrimental [25,26,27,31,32]. To conclude, our data show that IL11 is profibrotic and proinflammatory in the heart, and we propose IL11 as a potential therapeutic target for fibro-inflammatory heart disease.

## 4. Materials and Methods

### 4.1. Animal Studies

The mice were housed in the same room with a 12-h light–dark cycle and provided with food and water ad libitum. Female *Rosa26-Il11* mice on a C57BL/6N background were used as previously described (Cat: 031928) [5], and male MCM on a C57BL6/J background were purchased from Jax (Cat:005657).

Expression of *Il11* was induced in 8-week-old mice that were heterozygous for both MCM and *Rosa26-Il11* transgenes (designated as MCM^Cre/+^:R26R*^Il11^*^/+^; CM-Il11-Tg). Age-matched littermates that were heterozygous for the MCM gene and wild-type for the *Rosa26-Il11* transgene (MCM^Cre/+^:R26^+^^/+^; Controls) were used as control animals to account for the Cre-mediated toxicity phenotypes previously reported in MCM mice [39].

Tamoxifen (Sigma-Aldrich, St. Louis, MO, USA, T5648) was dissolved in ethanol (Fisher Scientific, Waltham, MA, USA, 10437341, 10% *v*/*v*) and corn oil (Sigma-Aldrich, C8267) and administered intraperitoneally for 5 consecutive days at 20 mg/kg from 8 weeks of age and followed-up at the designated time points (1, 3, and 6 weeks post-transgene induction).

### 4.2. Genotyping

The mice were bred in a dedicated breeding facility with ad libitum access to food and water with a controlled 12-h light–dark cycle. Genotype was confirmed using ear-notch DNA samples. The DNA was extracted using a sodium hydroxide digestion buffer, then diluted in 1M Tris-HCl, pH 8. The *Rosa26-Il11* transgene genotype was confirmed using a single PCR reaction, yielding a PCR product at 270 bp in the wild-type or 727 bp in the transgenic mouse. MCM mice were genotyped using two reactions for either the transgenic gene at 295 pb or the wild-type gene at 300 bp, along with an internal positive control. The primers used in these reactions are detailed in Appendix A.

### 4.3. Cardiomyocyte Extraction

Male wild-type C57BL/6J mice at 12 weeks of age were deeply anaesthetized with ketamine and xylazine before the heart was harvested. The chest was opened, and the heart and aorta were transferred to ice-cold perfusion buffer (113 mM NaCl, 4.7 mM KCl, 0.6 mM KH_2_PO_4_, 0.6 mM Na_2_HPO_4_, 1.2 mM MgSO_4_-7H_2_O, 12 mM NaHCO_3_, 10 mM KHCO_3_, HEPES Na Salt 0.922 mM, Taurine 30 mM, BDM 10 mM, glucose 5.5 mM, pH 7.4). Excess material was dissected away from the heart, and the ascending aorta was cannulated and placed on a retrograde perfusion Langendorff apparatus. The heart was perfused with warmed perfusion buffer for 4 min and then digestion buffer (0.2 mg/mL Liberase TM, 5.5 mM Trypsin, 0.005 U/mL DNase, 12.5 µM CaCl_2_) until fully digested. Subsequently, the heart was minced and gently pipetted until a uniform suspension was formed. The cells were filtered through sterile gauze and resuspended in perfusion buffer. Sequential calcium reintroduction was performed in 3 steps up to 1 mM (62 µM, 212 µM and 1 mM). Cardiomyocytes were plated on laminin and allowed to equilibrate for 2 h in M199 media supplemented with 2 mM L-carnitine, 5 mM creatine, and 5 mM taurine. Blebbistatin (Sigma Aldrich, 203390) was added at 25 µM to prevent cardiomyocyte contraction and extend culture life. The cells were treated with media containing recombinant mouse transforming growth factor-beta 1 (rmTGFβ1) at 5 ng/mL (R&D systems, Minneapolis, MN, USA, 7666-MB) or normal media. The media was removed after 24 h, the cells were washed, and the RNA was harvested using TRIzol reagent (Invitrogen, Carlsbad, CA, USA, 15596026).

### 4.4. Echocardiography

Echocardiographic images were acquired under anesthesia using a Vevo3100 imaging system (Version 5.5.0, Fujifilm Visualsonics Inc., Toronto, ON, Canada). The anesthesia was induced using 4% isoflurane and maintained with 1.5–2% isoflurane. Volumetric measurements were made using VevoLab software (Version 5.5.0, Fujifilm Visualsonics Inc.) from m-mode images in the parasternal long axis view and were averaged across three heartbeats. Strain analysis was performed in a semi-automated manner using VevoStrain function of VevoLab Software (Version 5.5.0, Fujifilm Visualsonics Inc.) software from parasternal short axis images.

### 4.5. qPCR

The heart was dissected from the chest and washed in ice-cold PBS and snap-frozen in liquid nitrogen. Total RNA was extracted using TRIzol (Invitrogen, 15596026) in RNeasy columns (Qiagen, Toronto, ON, Canada, 74106). cDNA was synthesized using Superscript Vilo Mastermix (Invitrogen, Carlsbad, CA, USA, 11755050). Gene expression analysis for *Il11* was performed using a Taqman probe over 40 cycles. The expression of genes involved in fibrosis and extracellular matrix turnover was analyzed in duplicate using a custom TaqMan array microfluidic card. A total of 100 µL of sample cDNA and master mix was added to each well, and the card was centrifuged twice for 1 min at 331× *g* and analyzed in duplicate over 40 cycles. The primers and probes used are detailed in Appendix A. The expression data were normalized to *Gapdh* mRNA expression, and the fold-change compared to the control samples was calculated using 2^−∆∆Ct^ method.

### 4.6. RNASeq

The RNA quality for RNAseq was assessed using the Agilent 2100 RNA 6000 Nano assay and RNASeq libraries prepared using the NEBNext Ultra II Directional RNA Library Prep Kit with NEBNext Poly(A) mRNA Magnetic Isolation Module following the manufacturer’s instructions.

The library quality was evaluated using the Agilent 2100 High-Sensitivity DNA assay, and the concentrations were measured using the Qubit™ dsDNA HS Assay Kit. The libraries were sequenced using a NextSeq 2000 to generate a minimum of 20 million paired-end 60 bp reads per sample.

The raw reads were aligned to the mouse reference genome GRCm39 (Ensembl release 106) using the STAR [40] software v2.7.10a with default parameters. The aligned reads were assigned to genes to produce a gene-level count matrix using the featureCounts tool [41] as part of the Rsubread package v2.10.4. Differential gene expression analysis was performed using the edgeR software [42] v3.38.1 following the author’s recommended analysis pipeline utilizing a quasi-likelihood negative binomial generalized log-linear model.

Gene set enrichment analysis was carried out on fold-change ranked gene lists from both male and female mice at 1, 3, and 6 weeks post-tamoxifen induction of *Il11* expression. The data was imputed into the GSEA software v4.3.2. (Broad Institute, San Diego, CA, USA and UC, San Diego, CA, USA) with MSigDB Hallmark gene sets [43,44].

KEGG pathway analysis was performed using a combined list of differentially expressed genes across all three time points. The gene lists were imputed into the ShinyGO v0.77 software (South Dakota State University, Brookings, SD, USA) [45], and the top 20 KEGG pathways ranked by enrichment false discovery rate (FDR) values were visualized.

### 4.7. Protein Analysis

Protein extraction was performed using ice-cold Pierce RIPA buffer (ThermoFisher, Waltham, MA, USA, 89901) supplemented with protease inhibitors (Roche, Rotkreuz, Switzerland, 11697498001) and phosphatase inhibitors (Roche, 4906845001). Tissue was lysed using a Tissue Lyser II (Qiagen, Venlo, The Netherlands) with metallic beads for 3 min at 30 Hz. Protein quantification was performed using a Pierce BCA colorimetric protein assay kit (ThermoFisher, 23225). A total of 10µg of protein was loaded per well and run on a 4–12% bis-tris precast SDS page gel (Invitrogen, NP0323BOX). Semi-dry transfer was performed using the TransBlot Turbo transfer system (BioRad, Hercules, CA, USA, 1704150), and the membrane was blocked in 5% BSA (Sigma-Aldrich, A3803). The membrane was incubated overnight at 4 °C with primary antibodies including IL11 (R&D Systems, mab218), phospho-STAT3-Tyr705 (Cell Signalling, Danvers, MA, USA, 9145S), STAT3 (Cell Signalling, 4904S), and TWIST (Abcam, Waltham, MA, USA, ab50887) raised in rabbit. The appropriate secondary HRP-linked antibody was incubated for 1 h at room temperature and developed using Clarity Western ECL blotting substrate (BioRad, 1705061).

Blood samples for circulating IL11 quantification were collected following cervical dislocation and centrifuged at 2000× *g* for 20 min at 4 °C. The serum fraction was then aspirated and frozen. IL11 serum quantification was performed using a mouse IL11 colorimetric sandwich ELISA kit (Abcam, ab214084) as per the manufacturer’s instructions with a 1:4 dilution and compared to the standard curve.

### 4.8. Masson Trichrome Staining

Heart sections were fixed overnight in 10% formalin (Sigma-Aldrich, F5554) and then in ethanol (Fisher Scientific, 10437341, 70% *v*/*v*). The samples were embedded in paraffin, and staining was performed using a Masson’s trichrome kit (Sigma, HT15-1KTKIT) according to the manufacturer’s protocol. Images for analysis were acquired using a Zeiss Axio scan digital slide scanner. The total global collagen was quantified using the color threshold function in the Fiji software (version 2.0.0) to identify fibrotic regions and compared to the whole myocardial area. Perivascular fibrosis was assessed as the degree of fibrosis surrounding the vessels compared to the vessel size. The five largest vessels in each section were analyzed, and an average was taken for each animal.

### 4.9. Immunohistochemistry

Immunohistochemical staining was performed on formalin-fixed, paraffin-embedded samples. The myocardial sections were deparaffinized and permeabilized with methanol, and antigen retrieval was performed using citrate buffer (Abcam, Ab64214, pH-6 at 100 °C). The slides were blocked in DAKO peroxidase blocking solution (Aglient Techniologies, Santa Clara, CA, USA, S2023) and serum-free blocker (Leica Biosystems, Deer Park, IL, USA, RE7102.) The sections were then incubated with primary antibody against MAC2 (Cedarlane, Burlington, ON, Canada, CL8942AP) overnight at 4 °C and secondary anti-rat (Vector, Stuttgart, Germany, BA-4001) for 1 h. The staining was visualized using a NovoLink polymer detection system (Leica Biosystems: RE7112, RE7105, RE7143). The sections were then counterstained with hematoxylin and imaged using a Zeiss Axio scan digital slide scanner. Quantification of immunhistochemical staining was performed using Qupath software 0.2.0 simple thresholder to quantify the percentage of staining in the whole myocardium or an area surrounding the vessels. Perivascular quantification was performed on the 3 largest vessels in each section, and an average was taken for each animal.

### 4.10. Statistical Analysis

All statistical analyses were performed in GraphPad Prism V9.5.0. Normality testing was performed using the Shapiro–Wilk test. Hypothesis testing for single comparisons was done using an unpaired two-way Student’s t test. Comparisons involving male and female mice were performed using a two-way analysis of variance (ANOVA, San Francisco, CA, USA) with Sidak’s multiple comparisons testing. Changes in expression over time were analyzed using a one-way ANOVA, and the changes across genotypes, sex, and time in Appendix A were tested using a 3-way ANOVA. All the graphs display the mean and standard error of the mean unless otherwise stated. The *p*-values in the RNA seq analysis were corrected for multiple testing using the false discovery rate method. A *p*-value and FDR of <0.05 was considered significant.

## Figures and Tables

**Figure 1 ijms-24-12989-f001:**
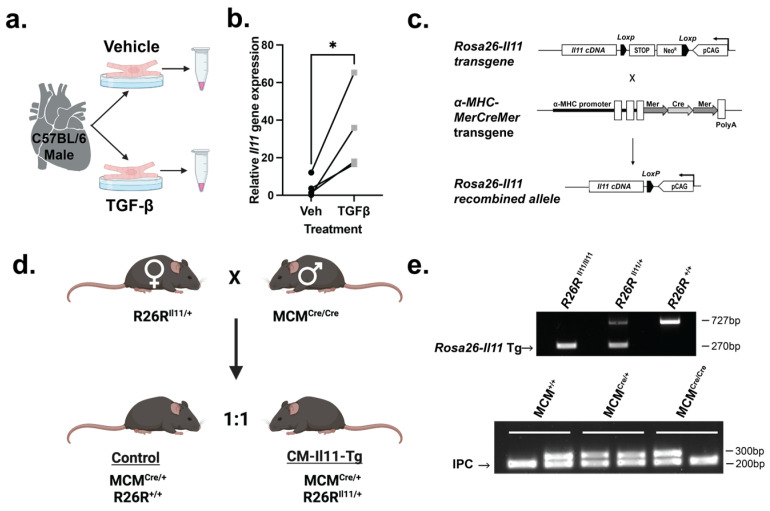
Interleukin 11 (*Il11*) expression in cardiomyocytes and generation of a new *Il11* transgenic mouse. (**a**) Schematic of cardiomyocyte extraction and transforming growth factor β (TGFβ) treatment protocol from wild-type C57BL6/J mice. (**b**) *Il11* mRNA levels using qPCR from cell lysates of isolated cardiomyocyte samples after 24 h in normal media (●) or media supplemented with TGFβ (■). Cardiomyocyte preparations isolated from the same heart with and without TGF-β are connected by lines (n = 4 vs. 4). (**c**) Schematic diagram of the targeted expression of *Il11* in cardiomyocytes. In the *Rosa26-Il11* transgene, a floxed cassette containing both the neomycin (neo) resistance and stop elements is positioned before the murine *Il11* transgene cassette, which undergoes tamoxifen-initiated, Cre-mediated recombination when crossed with the α-MHC-MerCreMer (MCM) mouse (adapted from Lim et al. [19]). (**d**) Breeding scheme used to produce CM-Il11-Tg mice and littermate controls. (**e**) Genotyping gels of ear notch biopsy DNA from control and experimental mice. A 270 bp band indicated the presence of the *Rosa26-Il11* transgene, and the wild-type band appeared at 727 bp (top gel). The MCM band is genotyped using 2 reactions with an internal positive control (IPC). Bands for the wild-type allele or Cre transgene are at 295 bp and 300 bp, respectively. The IPC band appears at 200 bp (bottom gel). Statistics: Two-tailed unpaired t test. Significance denoted as * *p* < 0.05.

**Figure 2 ijms-24-12989-f002:**
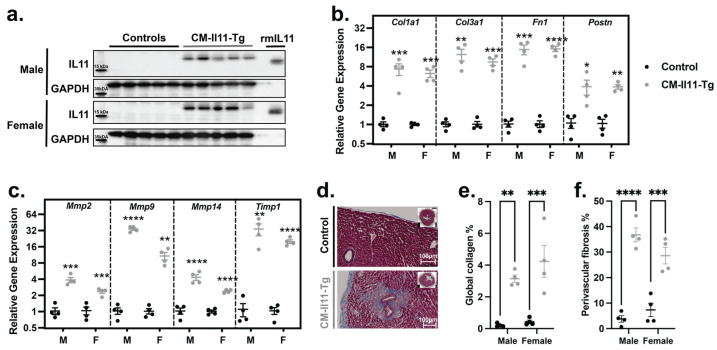
Cardiomyocyte-restricted IL11 expression causes myocardial fibrosis. (**a**) Ventricular expression of IL11 and GAPDH on a Western blot 6 weeks after induction of the transgene in male and female control and CM-Il11-Tg mice (n = 5 vs. 5 mice per sex). QPCR analysis of (**b**) fibrosis and (**c**) extracellular matrix remodeling genes in the ventricular myocardium of male (M) and female (F), control (●) and CM-Il11-Tg (●) mice (n = 4 vs. 4). (**d**) Representative Masson’s trichrome staining of male control and CM-Il11-Tg mice. Quantification of (**e**) global collagen content of myocardium and (**f**) relative perivascular collagen compared to vessel area (n = 4 vs. 4). Statistics: Two-way ANOVA with Sidak’s multiple comparisons test. Significance is denoted as * *p* < 0.05, ** *p* < 0.01, *** *p* < 0.001, **** *p* < 0.0001.

**Figure 3 ijms-24-12989-f003:**
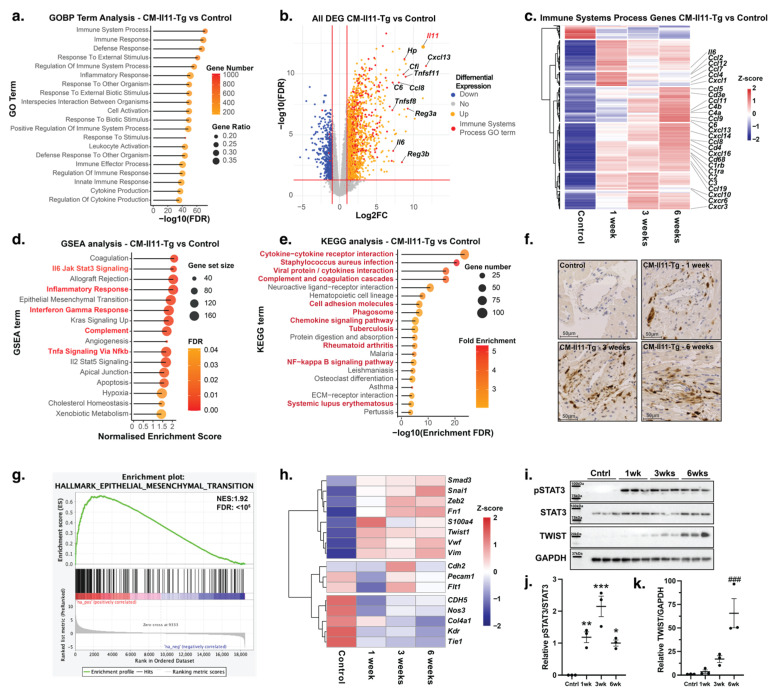
Bulk RNA-seq analysis of the transcriptional effects of IL11 expression in cardiomyocytes. (**a**) GO term analysis of RNAseq experiment from the ventricles of male CM-Il11-Tg mice (n = 4 per group) 6 weeks after transgene induction with tamoxifen compared to control animals. (**b**) Volcano plot of all differentially expressed genes (Up = 1789 genes, Down = 525 genes) in the ventricles from male CM-Il11-Tg mice 6 weeks post-transgene induction compared to control animals. Genes included in the most enriched GOBP term “immune system processes” term are highlighted in red (493/2790 genes), and the top 10 most differentially expressed genes in this GO term are labelled (n = 4 per group). (**c**) Heatmap with hierarchical clustering of z-scores of differentially expressed “immune system process” genes from the ventricular myocardium of male CM-Il11-Tg mice at 1, 3, and 6 weeks after transgene induction compared to control animals. Selected genes involved in leukocyte chemotaxis and the complement system are labelled (n = 4 per group). (**d**) Significantly enriched gene sets from GSEA of 6-week-old male CM-Il11-Tg mice compared to controls using MSigDB Hallmark gene sets (n = 4 per group). (**e**) Top 20 most significantly enriched KEGG terms in the pooled analysis of all differentially expressed genes across 3 time points in male CM-Il11-Tg mice compared to controls (n = 4 per group). KEGG terms involved in the inflammatory response are highlighted in red. (**f**) Immunohistochemical staining of macrophages using MAC2 antibody in the perivascular region of control mice and CM-Il11-Tg male mice 1, 3, and 6 weeks after transgene induction. (**g**) Enrichment plot of epithelial-to-mesenchymal transition (EMT) gene set (M5930) using Broad GSEA software (Version 4.3.2) 6 weeks after induction of the transgene in CM-Il11-Tg mice compared to control mice (n = 4) (NES:1.92, FDR: <0.0001) (**h**) Heatmap of z-scores of genes involved in endothelial-to-mesenchymal transition, including mesenchymal markers, endothelial markers, and related transcription factors at 1, 3, and 6 weeks post-tamoxifen induction compared to control animals. (**i**) Western blot and quantification of phosphorylated STAT3, total STAT3, and TWIST expression in the ventricular myocardium in control mice and CM-Il11-Tg mice 1, 3, and 6 weeks after induction of *Il11* expression (n = 3 per time point). Quantification of (**j**) pSTAT3/STAT3 and (**k**) TWIST/GAPDH Western blots from (**i**) (n = 3 per group) Statistics: One-way ANOVA with Sidak’s multiple comparisons test. Significance denoted as * *p* < 0.05, ** *p* < 0.01, *** *p* < 0.001, ### *p* < 0.001 compared to all other groups.

**Figure 4 ijms-24-12989-f004:**
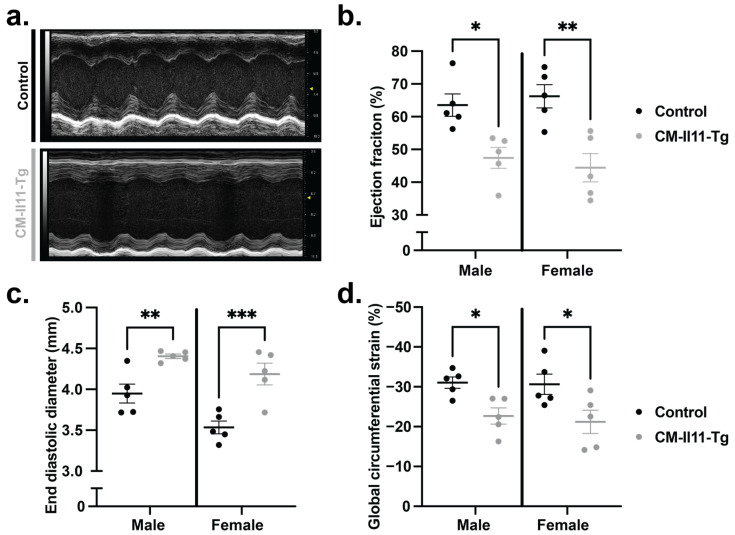
Cardiomyocyte IL11 expression causes left ventricular systolic impairment. (**a**) Representative parasternal long axis (PLAX) m-mode images from male control and CM-Il11-Tg mice 6 weeks after induction of *Il11* expression. (**b**) Quantification of left ventricular ejection fraction from PLAX m-mode images in male and female CM-Il11-Tg mice compared to control animals 6 weeks after transgene induction (n = 5 vs. 5 per sex). (**c**) End diastolic diameter measured from PLAX m-mode images (n = 5 vs. 5 per sex). (**d**) Global circumferential strain measured using Vevostrain speckle tracking software (VevoLab version 5.5.0) from the parasternal short axis images (n = 5 vs. 5 per sex). Statistics: Two-way ANOVA with Sidak’s multiple comparisons test. Significance is denoted as * *p* < 0.05, ** *p* < 0.01, *** *p* < 0.001.

## Data Availability

All data are provided in the manuscript and Appendix A. Raw RNAseq data and gene-level counts have been uploaded to the NCBI Gene Expression Omnibus database and will be made available using accession number (GSE236854).

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
