# Peer review of "Cardiomyocyte-Restricted Expression of IL11 Causes Cardiac Fibrosis, Inflammation, and Dysfunction"

_ijms, 2023, doi:10.3390/ijms241612989_

Round 1
Reviewer 1 Report
In the submitted manuscript, Sweeney and colleagues describe a novel transgenic mouse strain with cardiomyocyte restricted expression of IL-11, and analyze the consequences of the transgene for heart function & heart fibrosis. The most important point, namely the finding that IL-11 induces heart fibrosis in mice, has already been described previously by the same lab. The here reported data is a small addition to their previous work and the novelty is rather limited. Nevertheless, the experiments are thorough and support the conclusions that are drawn from the data. There are only minor issues that should be addressed:
In line 49 the authors claim that “In human studies, administration of exogenous IL11 causes heart failure symptoms” – The cited references, however, do not describe the administration of exogenous IL-11, but rather correlate blood levels of endogenous IL-11 with different heart symptoms and biomarkers. Is the reference missing or does a study evaluating the consequences of exogenous IL-11 in humans not exist?
There is a rather high amount of self-citations. While this might be unavoidable, as the authors have obviously done tremendous work linking IL-11 and fibrosis in recent years, I encourage them to double check whether i) all self-citations are strictly necessary and ii) they have cited all important work from other labs.
The western blots in Fig S1A and 2A both show IL-11 expression in the mouse heart with GAPDH as control. Surprisingly, the molecular weight of both proteins is labeled differently in the two blots. Why is that the case? The uncropped blots for both figures are missing as well, which are required here to understand the differences. Furthermore, in Fig 2A, why is rmIL-11 apparently smaller than the transgenic IL-11?
The authors state in line 43 that “IL11 is not easily detected in healthy tissues”. However, in Fig S1C the IL-11 serum concentration in the non-transgenic mice is up to 500 pg/ml – do the authors have any explanation for these apparently high concentrations?
The authors might consider to describe their results regarding ventricular function (section 2.5) in a little more detail. As the journal is rather targeted towards molecular biologists, the readers might not be very familiar with this type of data.
In the discussion (line 237f) the authors suggest that cardiomycocyte stress induces IL-11 secretion. Is there any evidence to base that suggestion on?
In the same paragraph, the authors claim that loss of IL-11 signaling in mice has no adverse effects (without offering any references). While the effects are obviously mild compared to loss of TGFb, mice still suffer from different bone abnormalities, and female mice are infertile. At least the bone abnormalities are also reported in humans, often accompanied by mental impairment. The evaluation of the safety profile of an anti-IL-11 therapy should consider these well known data.
Some small formatting/figure issues:
- line 166: references not formatted correctly
- Fig 3I: molecular weight of protein signals not labeled
- Fig 4A: both images are labeled identically
- Fig 4: Why is this the only figure were the ANOVA results are also written within the figure? Are the data in 4D significantly different or not? (because there are no asterisks)
Author Response
In the submitted manuscript, Sweeney and colleagues describe a novel transgenic mouse strain with cardiomyocyte restricted expression of IL-11, and analyze the consequences of the transgene for heart function & heart fibrosis. The most important point, namely the finding that IL-11 induces heart fibrosis in mice, has already been described previously by the same lab. The here reported data is a small addition to their previous work and the novelty is rather limited. Nevertheless, the experiments are thorough and support the conclusions that are drawn from the data. There are only minor issues that should be addressed:
In line 49 the authors claim that “In human studies, administration of exogenous IL11 causes heart failure symptoms” – The cited references, however, do not describe the administration of exogenous IL-11, but rather correlate blood levels of endogenous IL-11 with different heart symptoms and biomarkers. Is the reference missing or does a study evaluating the consequences of exogenous IL-11 in humans not exist?
Thank you for highlighting this omission. We have added the appropriate reference from Liu et al and have also added a second reference from Yu et al which describes myocardial hypertrophy and fluid retention in non-human primates following IL11 administration.
There is a rather high amount of self-citations. While this might be unavoidable, as the authors have obviously done tremendous work linking IL-11 and fibrosis in recent years, I encourage them to double check whether i) all self-citations are strictly necessary and ii) they have cited all important work from other labs.
(i) Thank you we have checked through citations and removed any redundant self-citations or replaced these with citations from other authors: The following citations were removed:
Ng B, Dong J, D’Agostino G, et al. “Interleukin-11 is a therapeutic target in idiopathic pulmonary fibrosis.” Sci Transl Med. 2019;11:511:eaaw1237. doi:10.1126/scitranslmed.aaw1237
Lim WW, Corden B, Ye L, et al. “Antibody-mediated neutralization of IL11 signalling reduces ERK activation and cardiac fibrosis in a mouse model of severe pressure overload.” Clin Exp Pharmacol Physiol. 2021;48:4:605-613. doi:10.1111/1440-1681.13458
Dong J, Viswanathan S, Adami E, et al. “Hepatocyte-specific IL11 cis-signalling drives lipotoxicity and underlies the transition from NAFLD to NASH.” Nat Commun. 2021;12:1:66. doi:10.1038/s41467-020-20303-z
- ii) We have increased the number of citations from other groups in the manuscript and reference to the following papers can now be found in the manuscript.
- Kapina MA, Shepelkova GS, Avdeenko VG, et al. (2011) Interleukin-11 drives early lung inflammation during Mycobacterium tuberculosis infection in genetically susceptible mice. PloS one 6(7): e21878.
- Seyedsadr M, Wang Y, Elzoheiry M, et al. (2023) IL-11 induces NLRP3 inflammasome activation in monocytes and inflammatory cell migration to the central nervous system. Proceedings of the National Academy of Sciences of the United States of America 120(26): e2221007120.
- Liu N-W, Huang X, Liu S, et al. (2019) Elevated BNP caused by recombinant human interleukin-11 treatment in patients with chemotherapy-induced thrombocytopenia. Supportive care in cancer: official journal of the Multinational Association of Supportive Care in Cancer 27(11): 4293–4298 .
- Yu K-M, Lau JY-N, Fok M, et al. (2018) Preclinical evaluation of the mono-PEGylated recombinant human interleukin-11 in cynomolgus monkeys. Toxicology and applied pharmacology 342: 39–49.
- Robb L, Li R, Hartley L, et al. (1998) Infertility in female mice lacking the receptor for interleukin 11 is due to a defective uterine response to implantation. Nature medicine 4(3): 303–308.
- Agthe M, Brügge J, Garbers Y, et al. (2018) Mutations in Craniosynostosis Patients Cause Defective Interleukin-11 Receptor Maturation and Drive Craniosynostosis-like Disease in Mice. Cell reports 25(1): 10-18.e5.
- Nieminen P, Morgan NV, Fenwick AL, et al. (2011) Inactivation of IL11 signaling causes craniosynostosis, delayed tooth eruption, and supernumerary teeth. American journal of human genetics 89(1): 67–81.
The western blots in Fig S1A and 2A both show IL-11 expression in the mouse heart with GAPDH as control. Surprisingly, the molecular weight of both proteins is labeled differently in the two blots. Why is that the case? The uncropped blots for both figures are missing as well, which are required here to understand the differences. Furthermore, in Fig 2A, why is rmIL-11 apparently smaller than the transgenic IL-11?
Thank you for highlighting the discrepancy. We have labelled the protein markers however different protein ladders were used in these blots. The supplementary figure was incorrectly labelled as 25kDa rather than 20kDa. These have been relabeled to give a consistent observed molecular weight of 19kDa which is the same as the theoretical molecular weight. All uncropped blots are available in the attached file.
The small discrepancy between endogenously produced IL11 and recombinant IL11 we suspect relates to post-translational modifications of the protein in vivo which do not occur when produced in vitro.
The authors state in line 43 that “IL11 is not easily detected in healthy tissues”. However, in Fig S1C the IL-11 serum concentration in the non-transgenic mice is up to 500 pg/ml – do the authors have any explanation for these apparently high concentrations?
We agree this is higher than expected and suspect this is related to the dynamic range of the ELISA assay at lower concentrations. We have repeated the analysis using available samples and a logarithmic standard curve. This result is more in keeping with the serum concentrations of IL11 previously reported in control mice WT (mean 99.4 pg/L) These values are close to the lowest standard concentration of the assay (78.12ng/L). Additionally, we performed a serum ELISA on control and CM-Il11-Tg mice used in the main study and found similar results for the control animals in both male and female mice (Male:148 pg/L, Female:82.8pg/L). This has also been included in the supplementary figure 1.
The authors might consider to describe their results regarding ventricular function (section 2.5) in a little more detail. As the journal is rather targeted towards molecular biologists, the readers might not be very familiar with this type of data.
Thank you for highlighting this we have included the following additions to the text to explain the measures of left ventricular ejection fraction and global circumferential strain.
The cardiac function was assessed by transthoracic echocardiography 6 weeks after transgene induction. Left ventricular ejection fraction (LVEF), which calculates the proportion of blood ejected during each cardiac contraction, had similar impairment in both male and female CM-Il11-Tg mice, as compared to tamoxifen injected controls (Fig 4a & b) suggesting the absence of sexual dimorphism of cardiomyocyte-specific Il11 expression on heart function. This was accompanied by dilation of the left ventricular cavity indicating adverse cardiac remodelling of the ventricle. Myocardial shortening, a complementary measures of systolic force generating capacity of the ventricle, was also reduced when measured using global circumferential strain (Fig 4 c & d).
In the discussion (line 237f) the authors suggest that cardiomycocyte stress induces IL-11 secretion. Is there any evidence to base that suggestion on?
We have altered this sentence to improve the clarity. Signaling molecules released in response to myocardial stress including TGF-B act on cardiomyocytes to stimulate Il11 expression as we have shown in Figure 1. Although IL11 is known to be increased in pressure overload and ischemia the cellular source of this release has not been defined and further work is needed to define this.
The sentence now reads: We suggest that myocardial stress signals that occurs with ischaemia or pressure overload [10,11] results in the secretion of IL11 from cardiomyocytes which leads to cardiac fibrosis and inflammation.
In the same paragraph, the authors claim that loss of IL-11 signaling in mice has no adverse effects (without offering any references). While the effects are obviously mild compared to loss of TGFb, mice still suffer from different bone abnormalities, and female mice are infertile. At least the bone abnormalities are also reported in humans, often accompanied by mental impairment. The evaluation of the safety profile of an anti-IL-11 therapy should consider these well known data.
Thank you, IL11 loss of function has a number of effects on fertility and on development including most markedly a craniosyostosis phenotype in both mice and humans. In the adult mouse blocking the IL11 pathway does not have significant detrimental effects even in longterm ageing studies which is ressuring with regards to its safety profile. We have included the following text in the manuscript.
IL11 has important physiological roles in development and fertility. Female Il11ra1 knockout animals are infertile [33]. Craniosynostosis is seen in up to 40% of Il11ra1 knockout mice which is also reported in humans with homozygous IL11RA loss of function mutations [34,35]. However, in the adult mouse, Il11 or Il11ra1 knockout mice or long-term anti-IL11 antibody experiments do not appear to cause major health issues indicating a favourable safety profile [36], again contrasting with genetic or pharmacologic inhibition of TGFβ that is detrimental [25,28].
Some small formatting/figure issues:
- line 166: references not formatted correctly
Thank you this has been corrected
- Fig 3I: molecular weight of protein signals not labeled
Thank you this has been corrected
- Fig 4A: both images are labeled identically
Thank you this has been changed to defined control and CM-Il11-Tg labels as previously defined in the methods section.
- Fig 4: Why is this the only figure were the ANOVA results are also written within the figure? Are the data in 4D significantly different or not? (because there are no asterisks)
In figure 4D although the ANOVA was positive for a trend both male and female were individually borderline significant p=0.07 and therefore did not have asterisks. We have in the interim added extra biological replicates which have strengthened the significance, and these have been changed on all graphs. We have also removed the ANOVA statistic which can be assumed to be positive given post-hoc tests are being undertaken and in keeping with other graphs in the manuscript.
Reviewer 2 Report
The authors examined the role of IL11 in cardiomyocytes in vivo. They showed that cardiomyocyte-specific IL11 overexpression exacerbates cardiac remodeling.
I have some comments.
1. Whether the effects of IL11 are pro-inflammatory or anti-inflammatory may be species specific, but may vary depending on the expression level/dosage, the expression period/duration of administration, and if it is a disease model, whether it is acute or chronic. So, the difference between the duration of administration of recombinant IL11 in the previous report and the duration of overexpression in this study, and whether it is for a disease model or not, should be clearly stated.
2. Since the authors have created mice that express IL11 specifically in cardiomyocytes, they should mention what changes occur in the cardiomyocytes themselves (e.g., histological or gene expression patterns).
3. about Figure 1b
Did authors compare the pre-treatment group with the post-treatment group? Or, compare the TGFβ non-treated group with the TGFβ-treated group? If authors compared the non-treated group with the treated group, it is strange to connect the dots. If authors compared the pre-treatment group with the post-treatment group, they cannot exclude the possibility that the 24-hour culture manipulation itself increased IL11 expression.
4. about Figure 3f
Images show inflammatory cell infiltration in the peri-arterial area. Were the inflammatory cells infiltrated locally around the artery? Or were they diffusely infiltrating other areas as well?
5. Authors mention changes in gene expression related to EMT/EndoMT and peri-arterial fibrosis, but is there any difference in capillary density?
Author Response
The authors examined the role of IL11 in cardiomyocytes in vivo. They showed that cardiomyocyte-specific IL11 overexpression exacerbates cardiac remodeling.
I have some comments.
- Whether the effects of IL11 are pro-inflammatory or anti-inflammatory may be species specific, but may vary depending on the expression level/dosage, the expression period/duration of administration, and if it is a disease model, whether it is acute or chronic. So, the difference between the duration of administration of recombinant IL11 in the previous report and the duration of overexpression in this study, and whether it is for a disease model or not, should be clearly stated.
Thank you, these important details regarding previous studies concluding an anti-inflammatory effect have been added to the discussion and referenced. We have also compared this to changes seen in our model. The additional text below has been included:
The higher level of expression or more chronic exposure to IL11 in this model may be, in part, responsible for differences in inflammatory response, as compared to earlier mouse studies which had concluded an anti-inflammatory role of rhIL11. These studies ad-ministered rhIL11 to mice by daily injection and reported reduced pancreatic islet de-struction in type 1 diabetes models, reduced synovitis in rheumatoid arthritis models and reduced colitis in an inflammatory bowel disease model [28-30]. The doses used in these studies were variable, however in some cases very large doses of up to 100mcg per mouse were used in these studies making this possibility of a differential dose effect less likely. The duration of administration of rhIL11 was between 5 and 10 days of daily injection which is comparable to the 7 day RNAseq time point in our study by which point extensive inflammatory transcriptional changes are already present.
- Since the authors have created mice that express IL11 specifically in cardiomyocytes, they should mention what changes occur in the cardiomyocytes themselves (e.g., histological or gene expression patterns).
We have looked for changes in cardiomyocyte cell size using wheat germ agglutinin staining however did not see any differences in this measure in this model. Nppa and Nppb were upregulated in the RNAseq data particularly at the 1 week timepoint. We have included graphs of Nppa and Nppb fold change from RNA seq data in Fig S7d&e. The following text has been added to explain this
In keeping with this early effect on cardiac function cardiomyocyte stress markers Nppb and Nppa are significantly upregulated in the RNA-seq data by the 1-week timepoint in CM-Il11-Tg mice compared to WT mice [Fig S7d & e].
- about Figure 1b
Did authors compare the pre-treatment group with the post-treatment group? Or, compare the TGFβ non-treated group with the TGFβ-treated group? If authors compared the non-treated group with the treated group, it is strange to connect the dots. If authors compared the pre-treatment group with the post-treatment group, they cannot exclude the possibility that the 24-hour culture manipulation itself increased IL11 expression.
Thank you for highlighting this lack of clarity in the description of this experiment. In this experiment cardiomyocytes from each heart were split into 2 plates and we compared untreated to treated cells from the same hear after 24 hours in culture. The lines connect the individual preparations with and without TGF-B as significant variation exists within the cardiomyocyte extraction preparation protocol. An extra line has been added to the figure legend to explain the connecting lines.
(b) Il11 mRNA levels by qPCR from cell lysates of isolated cardiomyocyte samples after 24 hours in normal media (●) or media supplemented with TGFβ (■)(n=4vs4). Cardiomyocyte preparations isolated from the same heart with and without TGF-β are connected by lines
- about Figure 3f
Images show inflammatory cell infiltration in the peri-arterial area. Were the inflammatory cells infiltrated locally around the artery? Or were they diffusely infiltrating other areas as well?
The infiltration of MAC2 positive cells was predominantly in the perivascular region with small only small amounts of diffuse infiltration of cells. This has been highlighted in supplemental figures S6a&b which quantifies the MAC2 staining globally and in the perivascular region. This demonstrates the main significant difference is due to perivascular staining. We have included explanation of this in the text.
- Authors mention changes in gene expression related to EMT/EndoMT and peri-arterial fibrosis, but is there any difference in capillary density?
We have not directly checked for differences in capillary density in this model. Further work including capillary density and lineage tracking studies will be useful to further clarify the EndMT/EMT phenotype which has been raised by this study and will form part of ongoing work.
Reviewer 3 Report
Sweeney et al. investigated the roles of IL11 expressed in cardiomyocytes by establishing cardiomyocyte-restricted IL11 overexpressing mice and demonstrated that IL-11 plays deteriorative effects on cardiac function by causing inflammation and fibrosis. This study challenged important clarification of IL11, which has not been concluded. The methods of their approach and the results were very clear, and this manuscript provides novel and significant insights for patients with heart diseases in the future. However, there are some concerns before acceptation for publication in Int J Mol Sci.
1) Figure 3-f needs quantification.
2) Page 5 Line 157: “Fig 4e” must be a mistake. It should be Fig 3e.
3) Page 5 Line 166: “6.26” is unclear description.
Author Response
Sweeney et al. investigated the roles of IL11 expressed in cardiomyocytes by establishing cardiomyocyte-restricted IL11 overexpressing mice and demonstrated that IL-11 plays deteriorative effects on cardiac function by causing inflammation and fibrosis. This study challenged important clarification of IL11, which has not been concluded. The methods of their approach and the results were very clear, and this manuscript provides novel and significant insights for patients with heart diseases in the future. However, there are some concerns before acceptation for publication in Int J Mol Sci.
1) Figure 3-f needs quantification.
Thank you for highlighting this. The quantification for figure 3f has been placed in supplementary figure S6 to demonstrate small increase in global MAC2 staining and supplementary figure S6a which is predominantly caused by increased perivascular MAC2 staining displayed in S6b. Additionally we have included a graph of fold change of macrophage marker Cd68 from the RNA-seq experiment which confirms increase in macrophage expression within the myocardium.
2) Page 5 Line 157: “Fig 4e” must be a mistake. It should be Fig 3e.
Thank you this has been corrected to Fig3e
3) Page 5 Line 166: “6.26” is unclear description.
Thank you for highlighting this formatting error of the references call out, this has been placed inside brackets.